# Preliminary Ultrasonographic Study of Healthy California Sea Lion (*Zalophus californianus*) Pregnancy and Fetal Development

**DOI:** 10.3390/ani14091384

**Published:** 2024-05-05

**Authors:** Letizia Fiorucci, Francesco Grande, Roberto Macrelli, Pietro Saviano

**Affiliations:** 1Jungle Park, Urb. Las Aguilas del Teide s/n, 38640 Arona, Tenerife, Canary Islands, Spain; 2Loro Parque Fundación, Av. Loro Parque, s/n, 38400 Puerto de la Cruz, Tenerife, Canary Islands, Spain; 3DISPeA Department, University of Urbino, C. Bo, Via Piazza della Repubblica, 13, 61029 Urbino, Italy; roberto.macrelli@uniurb.it; 4Ambulatorio Veterinario Saviano-Larocca, 41042 Spezzano, Italy; drpietro@hotmail.it

**Keywords:** ultrasonography, California sea lion (*Zalophus californianus*), pregnancy, growth curve, fetal development, fetal welfare

## Abstract

**Simple Summary:**

Ultrasonography has been widely used in the field of veterinary medicine as a diagnostic tool for all kinds of animals, especially as a method of early diagnoses of pregnancy. Ultrasonography plays an important role in marine mammal preventative medicine and it is useful to monitor the estrus cycle, abnormal pregnancies, embryonic resorption, or fetal abortion. A large amount of studies is available on normal appearance during the fetal development in most domestic species; however, there is a lack of data to facilitate optimum diagnostic possibilities in marine mammals. This study provides additional information on maternal ultrasonographic monitoring and fetus wellness in California sea lion (*Zalophus californianus*) species, describing for the first time ultrasonographic findings from the ovulation up to the birth and the embryo and fetal organogenesis.

**Abstract:**

Reproductive success is an important aspect of marine mammals’ population health, as it is an indicator of the trajectory for the population into the future. The aim of this study is to provide additional relevant data on fetus–maternal ultrasonographic monitoring in sea lion species, in order to evaluate possible fetal distress or abnormalities. From 2018 to 2023, serial ultrasonographic scans of two healthy California sea lion females (16 ± 4 years old), kept under human care, were performed over the course of two pregnancies for each female. Animals were monitored from the ovulation to the delivery. Ultrasonography was performed weekly, and, during the last month, daily images were recorded using Logiq Versana Active, General Electric, with a 2–5 MHz curvilinear transducer, and Logiq V2, General Electric, with a 2–5 MHz curvilinear transducer. Right and left lateral recumbencies have been used during the examination. To the author’s knowledge, this is the first study describing in detail the sea lion organogenesis and their correlation with the stage of pregnancy.

## 1. Introduction

The reproductive cycle of pinnipeds consists of three basic phases: estrus, embryonic diapause, and fetal growth and development. Otariids generally have a postpartum estrus 6 to 12 days after delivery, the Californian sea lion being an exception, with this one occurring 1 month after birth. In phocids, estrus begins toward the end of lactation, which is much shorter (about 21–42 days) than in Otariids (6–12 months), or after weaning. Estrus can last from 1 to 9 weeks, with some animals being induced ovulators. Mating after a 28-day estrus yields a full-year reproductive cycle, resulting in an 11-month pregnancy with an approximate 90-day embryonic diapause [1]. Fertilization leads to embryonic cleavages until the early blastocyst stage [2]. Implantation is delayed until a photoperiod trigger affects circadian secretory patterns of prolactin and melatonin [1]. Diapause release leads to trophoblast implantation and the resumption of embryonic growth. Term development occurs around 242 days from the release of diapause [1,2]. Pinnipeds are classified as having obligate embryonic diapause. The time when the embryo resumes cellular divisions is a critical point during embryonic development of the fetus and, in non-pregnant females, is a period of the reactivation of sexual activity [1,2]. Pregnancy in pinnipeds can be divided into five distinctly important events: (1) conception, (2) embryonic diapause, (3) embryo reactivation and implantation, (4) fetal development, and (5) parturition [1]. In otariid species, it appears that an obligate pseudo-pregnancy ensues after ovulation, regardless of the presence of a normal blastocyst. However, after the 3 months physiologically allotted for embryonic diapause, uterine development and placental formation can only occur if a functional blastocyst is present [1]. During early post-conception, the embryo divides at a normal (compared with mammals without diapause) rate until the blastocyst stage around day 5 to 8 [1]. At this point, cellular divisions, as determined by a mitotic index, decline rapidly to a point where the embryo doubles in cell numbers every 50 to 60 days [1]. The embryo remains in this slow period of growth for 3 months until it is reactivated, and during this slow-growth period, the embryo remains in its zona pellucid. The reactivation of the blastocyst appears to be controlled by the photoperiod [1]. Water temperature and nutritional availability may also be important factors regulating pinniped reproductive cycles [1]. Implantation occurs in the midsection of the uterine horn, ipsilateral to the ovary that released the egg [1]. The diapause is the longest phase of pregnancy and it could be compared with the ovum phase of a dog [3]. After the fertilization of the ovum in the uterine tube, the zygote begins to divide rapidly. Canine zygotes take a longer interval to reach the utero-tubal junction than other species [3].

An ultrasound diagnosis of pregnancy has been used successfully in pinnipeds, but it is not able to detect the conceptus during embryonic diapause. Elevated progesterone levels are useful indicators of pregnancy, although high levels can also be found during pseudo-pregnancy [1]. 

Ultrasonography has been widely used in the field of veterinary medicine as an invaluable diagnostic tool for all kinds of animals, especially as a method of early diagnoses of pregnancy [4,5,6,7,8,9,10,11,12,13,14,15,16,17,18]. A thorough ultrasonographic evaluation of the reproductive tract should routinely be performed in sexually mature sea lion females, within a modern preventative medicine program using medical behaviors. The identification and monitoring of ovarian corpora, developing follicles, and corpus luteum (CL) establishes a normal baseline for each animal, detects reproductive cycle changes, and facilitates the diagnosis of pregnancy and pathology. A large amount of studies is available on normal appearance during the fetal development in most domestic species; however, there is a lack of data to facilitate optimum diagnostic possibilities in marine mammals [3]. In dolphins, transcutaneous scanning of pregnancy has been reported from day 78 to the end of gestation, from the 1988 study by Du Boulay and Wilson, and the organogenesis was described in detail by Saviano et al. in 2020 from day 29 to the delivery [4]. Lacave, for the first time, established the expected delivery date by the prediction from ultrasonographic measurements in two species of bottlenose dolphin (*Tursiops truncatus* and *Tursiops aduncus*) [19]. Most of the data collected on the reproduction of marine mammals come from by-caught or stranded animals and are therefore opportunistic in nature. Animals kept in a human-controlled environment offer a unique opportunity to gather data on the same individual with a known history over a long period of time. 

Reproductive success is vital in sustaining free-ranging and pinnipeds under human care populations, as it is an indicator of the trajectory for the population into the future [15,16,17,18]. California sea lions (*Zalophus californianus*) are exposed to some of the highest levels of contaminants worldwide because of their geographical range and trophic position. During the early 1970s, the high blubber levels of Dichlorodiphenyltrichloroethane (DDTs) and Polychlorinated biphenyl (PCBs) in California sea lions from San Miguel Island (CA, USA) were associated with premature pupping, stillbirths, and other reproductive failures [18]. To better understand these findings, promptly diagnose congenital anomalies, and enhance the early detection of pregnancy complications, there is a need for more sophisticated assessments of fetus–maternal health. 

In this study, authors used ultrasonography (US) to carry out a detailed observation of the embryonic and fetal growth process in pregnant California sea lion females under human care, considering that the mating dates have been known, in order to establish a relationship between the embryonic and fetal development and the gestation age and to evaluate possible disorders or abnormalities. However, ultrasonography has some limitations in this species, as the training in the population under human care, or the pharmacological or physical restraint for the free-ranging ones. Thus, this is a preliminary study, focused on the description of the ultrasonographic findings during the sea lion pregnancy, that produces better understanding of how the US investigation could be the diagnostic tool of choice for evaluating feto-maternal well-being in this wild species as it has been carried out previously in domestic species.

## 2. Materials and Methods

### 2.1. Study Animals

From 2018 to 2023, serial ultrasonographic scans of two healthy California sea lion females (16 ± 4 years old) were performed over the course of two pregnancies for each female. All examined animals were trained routinely for medical behaviors. Ultrasonography (US) was performed based on voluntary behavior in both animals kept under human care. California sea lion females were monitored from the ovulation to the delivery. Right and left lateral recumbencies have been used during the examination. Each animal was trained to maintain the required position for approximately 15 min. For the purposes of this study, the animal was trained to rest on its right side, exposing its belly on the left side and vice versa. Lateral and ventral abdominal scanning was performed using acoustic gel for acoustic coupling. Animals were not shaved. 

### 2.2. Ultrasonography: Instrumentation and Methodology 

Ultrasonography (US) was performed weekly, and, during the last month, daily. Images were recorded using Logiq Versana Active, General Electric, with a 2–5 MHz curvilinear transducer, and Logiq V2, General Electric, with a 3–5 MHz curvilinear transducer. The indicator of the probe was always direct to the head of the sea lion female. During the exam, a dark-colored bag was used to cover the instrument, to avoid direct sunlight. Some US exams were performed in a shaded closed area, to minimize the direct sunlight. A total of more than 200 ultrasound exams were included in this study. The calves born were 1 male and 3 females; of these, 1 female died 199 days after conception. For this study, the urogenital area was explored and the reproductive organs, the ovary and uterus, evaluated, describing their morphological changing during the pregnancy stages. During the embryonic and fetal development, each organ was recognized, recorded, and measured, in order to establish a possible relationship with the stage of pregnancy. Umbilical cord vasculature was assessed in cross-sections. Fetus heart rate frequency (HRF) was recorded as soon as it was possible to identify. Color Pulse Doppler confirmed vascular flow. It was possible to measure the bi-parietal diameters of the fetus as soon as it was possible to distinguish the head from the rest of the body. The head is visible as a symmetrical ovoid, showing the midline echo of the falx between the parietal bones, and it was measured at the widest section. For the thoracic diameter, measurements were taken at the level where the four chambers of the heart, surrounded symmetrically by the lungs, were visible. The pectoral flippers are often visible in the same section. Due to the fetal position, it was not possible to obtain a fair number of thoracic diameters, as it has been previously conducted with dogs and bottlenose dolphins [4,19,20]. These measurements were taken as previously carried out in bottlenose dolphins by Lacave and then used for the development of the growth models [4,19,20]. Moreover, fetal blubber thickness was measured in order to establish a relationship with the gestational stages. The measurements were made as soon as possible after conception. Finally, fetal position was evaluated during all the gestational periods.

### 2.3. Statistical Analyses

The main goal of this study was to model the expected value of the diameter of the fetal head (the output variable) in terms of the gestation age (the exposure variable), the blubber thickness (the output variable) in terms of the gestation age (the exposure variable), and the diameter of the fetal head (the output variable) in relation to the blubber length (the exposure variable).

Regression curves for these couple of variables were made using Medcalc software, Version 11.6.0.0, to analyze the data. A set of statistical processes for estimating the regression parameters and F-ratio significance level were determined.

## 3. Results

### 3.1. Changes in the Maternal Reproductive System

In the ultrasonography scan (US), a sea lion’s ovary appears caudally to the kidney. Considering the four pregnancies, the percentage of ovulation in the left ovary was 75%, while the ovulation in the right ovary was 25%. The Maximum Corpus luteum (CL) longitudinal diameter was 2.03 cm (Figure 1).

### 3.2. Relationship between Ultrasonographic Image Findings of Embryonic and Fetal Growth and the Age of Gestation

Day 0 of gestation: It was assumed to be the day of mating that corresponds to the day of ovulation.Day 113 ± 4 post-ovulation: The embryonic vesicle was first recognizable in the uterine cavity as a roundish structure with an average diameter of 1.71 cm with an anechoic content. In addition, it is possible to recognize the embryo inside it as an elongated hyperechoic structure (Figure 2). Thanks to these findings, it has been possible to make a diagnosis of pregnancy.

**Figure 2 animals-14-01384-f002:**
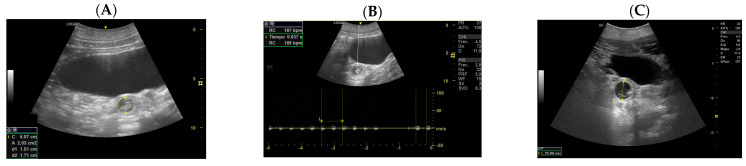
(**A**) The embryonic vesicle at 113 ± 4 days post-ovulation appears as a hyperechoic roundish structure with an average diameter of 1.5 cm with an anechoic content and a hyperechoic structure inside, ventrally to the urinary bladder. (**B**) It is possible to recognize from the beginning of the heart rate (189 bpm in the image). (**C**) The embryonic vesicle at 137 ± 2 days post-ovulation reached 2.56 cm in diameter.

Day 129 ± 3: Embryonic cardiac mechanics is displayed, as a point of maximum fluctuation of the echoes. It is possible to recognize from the beginning of the heart rate. The heart rate was measured once the cardiac mechanics became visible and remained constant between 205 and 155 bpm until the ninth month of pregnancy. For the next 3 months, it stabilized at 140–135 bpm (Figure 3).

**Figure 3 animals-14-01384-f003:**
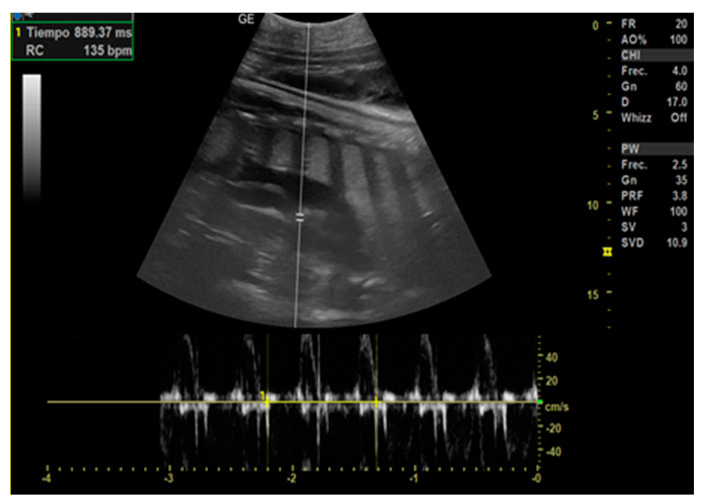
Around 129 ± 3 days after ovulation, the heart (H) is recognizable as an anechoic cavity and the embryonic cardiac mechanics is displayed, as a point of maximum fluctuation of the echoes. The heart rate measured one month before birth showed a frequency of 135 bpm.

Day 162 ± 2 and 170 ± 2: Skeletal formations, such as the cranial bone, vertebrae, costal bones, limbs, and detailed regions such as fingers, were first observed. The first abdominal organs to be visualized are the stomach and the urinary bladder, which appear as distinct and anechoic cavities. At the same time, it was possible to observe fetal movements.Day 186 ± 2: It is possible to identify the heart, lungs in the thorax, diaphragm in the middle, and liver and stomach in the abdomen. (D) Lungs appear as hyperechoic structures on the sides of the heart, and the latter seems to have a similar echogenicity compared to the liver (Figure 4).

**Figure 4 animals-14-01384-f004:**
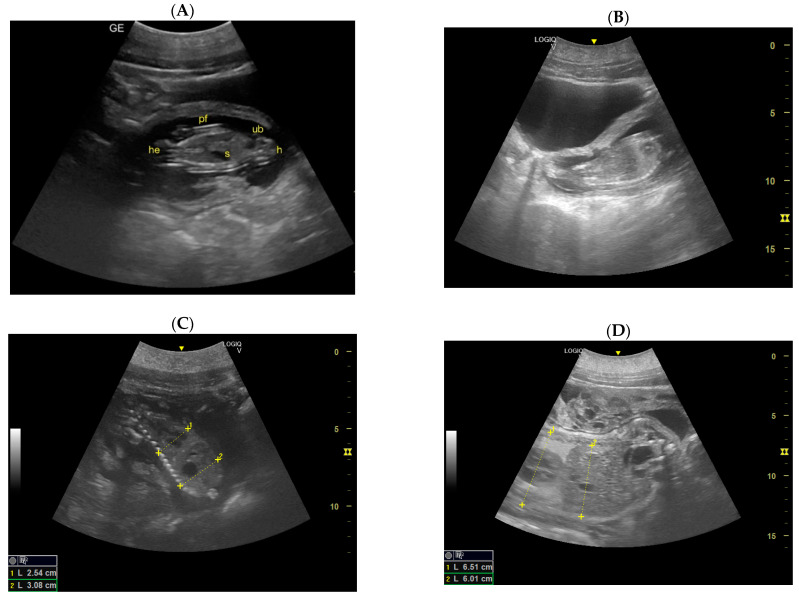
(**A**) The fetal head (he), fetal hip (h), fetal stomach (S), and fetal urinary bladder (ub) are the first abdominal organs to be visualized, and they appear as distinct and anechoic cavities; (**B**) heart chambers are visible in the middle of the fetal thorax; (**C**) it is possible to identify the lungs and liver separated by the diaphragm; (**D**) lungs appear as hyperechoic structures on the sides of the heart, and the latter seems to have a similar echogenicity compared to the liver.

Day 197: The umbilical cord is already seen like a hyperechoic cord form structure; it is important to identify the course, and to evaluate the internal vascular components and the absence of knots or torsions until the birth (Figure 5).Day 219 ± 3 days: It is also possible to recognize the eye as an anechoic cavitary structure (Figure 6). During the last month of pregnancy, it is possible to easily identify most of the abdominal organs such as the liver, spleen, intestine, and kidneys (Figure 7). In addition, the genitalia are visible and it could be possible to establish the sex of the fetus: in males, it is possible to identify the penis bone, which is obviously absent in females, but its visualization depends strongly on fetal position.

**Figure 5 animals-14-01384-f005:**
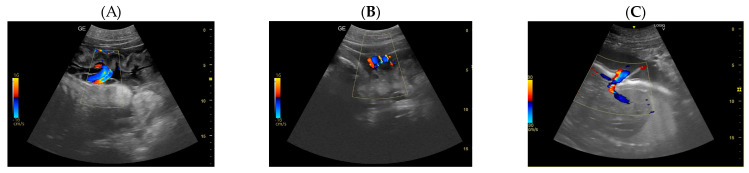
(**A**) The umbilical cord is already seen from the 197th day of gestation, and it appears like a hyperechoic cord form structure. (**B**) Color Pulse Doppler demonstrating flow within the umbilical vasculature: (**C**) the umbilical veins (in blue) and the umbilical arteries (in red).

**Figure 6 animals-14-01384-f006:**
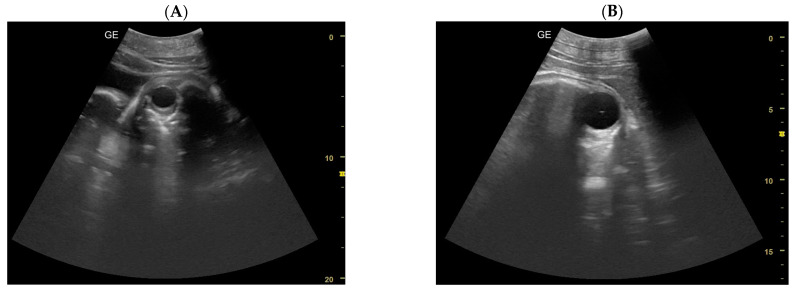
(**A**) From 219 ± 3 days of gestation, the eye is recognized as an anechoic cavitary structure; (**B**) starting from the 230th day, the lens is also visible and eyelid movements are also detectable.

**Figure 7 animals-14-01384-f007:**
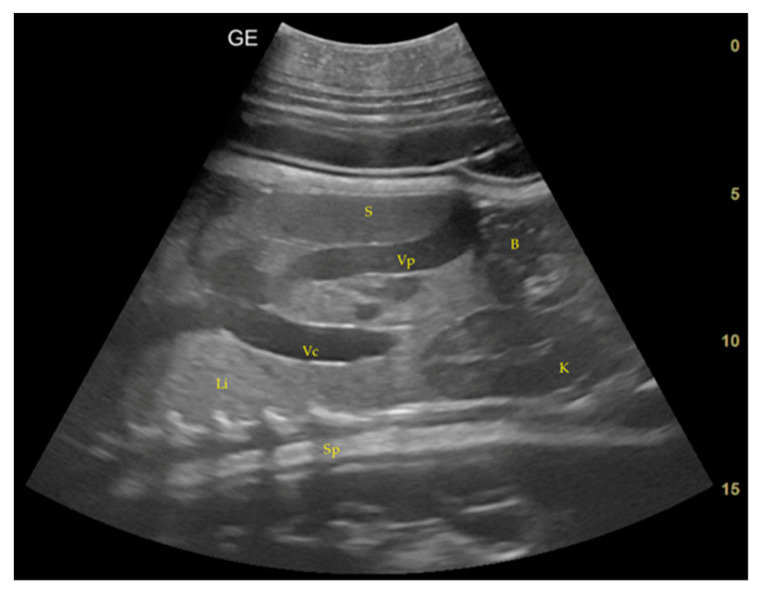
The fetal spinal cord (Sp), spleen (S), liver (Li), bowel (B), and kidney (K) and the vascular structures of the abdominal aorta, vena porta (Vp), and caudal vena cava (Vc) are visualized at 194 ± 5 days of gestation. The femurs, the tibias, the skull, and the ribs are visible as hyperechoic structures.

The number of days from the mating to the birth was 349 ± 6.

Allantoic and amniotic fluid were distinguishable in all examinations starting from the last trimester of pregnancy. Allantoic fluid appeared as an anechoic fluid, and the amniotic fluid appeared as a hyperechoic fluid, with an increasing amount of echoic particles during the last period of pregnancy.

Fetal position was evaluated during all the gestational periods. Regarding the four pregnancies, only one was a podalic presentation, while the other three calves were born with a head presentation. In the wild, approximately 63% of pups are delivered head-first [18]. It is interesting to note that it was a successful podalic delivery. According to the results of the present study, it is possible to predict the calf presentation at birth, considering its position in the uterus during the last trimester.

### 3.3. Construction of the Growth Curve of the Embryonic and Fetal Blubber Thickness

Femur and kidney measurements were made, as well as the blubber thickness, to evaluate the weighted growth of the fetus (Figure 8).

A growth curve of the embryonic and fetal blubber thickness was realized to monitor the normal trend of weight acquisition. Unfortunately, it was not possible to establish a fetal age on femur length as in humans [20] due to the fetal position that did not allow a sufficient number of data values.

### 3.4. Construction of the Growth Curve of the Embryonic and Fetal Head Diameter

Authors considered and realized the growth curve of the fetal head diameter to describe a curve of correlation between fetal head diameter and gestation age. The major and minor axes of the maximum cross-section of the embryonic and fetal heads were considered. The diameter of the fetal head was calculated as half the value of the sum of these two axes [19] (Figure 9). From these results, authors estimated the following linear regression equation of the growth curve between the gestation age (the independent variable X) and the diameter of the fetal head (the dependent variable Y) (Figure 10):Y = 0.5882X − 86.4151(1)
(R2 = 0.9508, F-ratio significance level *p* < 0.0001).

This growth curve shows that from around day 200 of gestation, the head grows at an almost fixed rate until day 335. After day 335, the rate sharply increases.

### 3.5. Construction of the Growth Curve of the Blubber Thickness

The following second-order regression equation of the growth curve between the gestation age (X) and the blubber thickness (Y) was obtained (Figure 11).
Y = 0.0014 × 2 − 0.7378X + 104.35(2)
(R2 = 0.8127, F-ratio significance level *p* = 0.0002).

This growth curve shows that from around day 250 of gestation, the blubber thickness grows at an almost fixed rate until day 335. After day 335, the rate sharply increases.

Furthermore, the blubber length and the diameter of the embryonic or fetal head are positively correlated (R2 = 0.9502, F-ratio significance level *p* < 0.0001) (Figure 12).

## 4. Discussion

In the present study, the authors describe by several US scans the fetal development, the fetal organs and structures, and their appearance, as described in other species [3,4,5,6,7,8,9,10,11,12,13,14,19,20,21,22,23]. 

The calves born were one male and three females; of these, one female died 199 days after conception. By US monitoring, the authors were able to detect an altered/slowed down growth, loss of fetal fluids and alteration of its shape, absence of the heartbeat, and blurring of margins and alteration of normal fetal anatomy, findings compatible with the death of the fetus. Consequently, the mother was treated properly with good results. A prompt diagnosis of fetal distress is essential to preserve mothers’ health when animals under human care are involved.

The growth curve of the embryonic and fetal head diameter was calculated, as previously described in other species [19,21]. This growth curve shows that from around day 200 of gestation, the head grows at an almost fixed rate. In addition, a growth curve of the embryonic and fetal blubber thickness was realized to monitor the normal trend of weight acquisition. Blubber, the lipid-rich hypodermis of marine mammals, serves several functions including thermal insulation, streamlining, buoyancy control, and a fuel and metabolic water source [24]. Blubber thickness is increased during postnatal growth. In emaciated adults, lipid mobilization is localized to the middle and deep blubber region. Thus, in terms of both lipid accumulation and depletion, the middle and deep blubber appear to be the most metabolically dynamic [24]. The superficial blubber likely serves a structural role important in streamlining the animal. A normal blubber thickness in a newborn is essential for its survival in the wild [24].

The pinnipeds’ placenta forms a zonary, an annular lobed band, similar to that of terrestrial carnivores, with labyrinthine arrangements [25]. A distinctive feature of a carnivore placenta is the presence of hemophagous regions of variable size and location, being the gross form very characteristic of the species [25]. These accessory structures constitute specialized areas of extravasated maternal blood, generally stagnant, surrounded by phagocytic trophoblasts. The hemophagous organ is the primary source of iron for the fetus [25]. In the domestic carnivores, the development of the conceptus, fetal membranes, and placentation has extensively been studied in cats and dogs [26]. Allantoic and amniotic fluid were distinguishable in all examinations starting from the last trimester of pregnancy. As occurring in dolphins, allantoic fluid appeared as an anechoic fluid, and the amniotic fluid appeared as a hyperechoic fluid, with an increasing amount of echoic particles during the last period of pregnancy [4].

Despite the large amount of studies regarding domestic species, there is a lack of information about pinniped, such as seals and sea lions, pregnancy and fetal development. Recent studies have described the embryogenesis and its relationship to the gestational age in dolphins [4]. However, to date, there have been no studies that describe the ultrasonographic evaluation of fetal development in pinnipeds. To the authors’ knowledge, this is the first study that reports sonographic descriptive findings of the California sea lion (*Zalophus californianus*) organogenesis and their correlation with the stage of pregnancy. 

This study provides additional data on maternal ultrasonographic monitoring and fetus wellness in this species. However, it showed some limitations as a low number of individuals and pregnancies. Moreover, the behavioral cooperation of the female and position of the fetus during any given exam dictated the quality/quantity of data acquired. In addition, the retrospective nature of an image analysis despite prospective image acquisition necessarily resulted in dataset truncation. Not all measurements could be derived from each set of captured cine and still images. Further studies are needed to measure the biometric parameters in order to predict the time of delivery by ultrasound, as previously described in other species [19,20,21,22,23]. There are several methods to estimate gestational age in the bitch and queen including ovulation timing, radiography, and ultrasonography [22,23]. Ultrasound can be used to monitor organ development, determine fetal viability, and assess fetal stress, uterine compromise, or placental disease [22,23]. Fetal stress can be assessed by examining the fetal fluids and fetoplacental units [26]. In dogs, previous studies confirmed a relationship between gestational age and fetal kidney growth [27,28]. Measurement of fetal kidney length can be used in conjunction with other methods to estimate gestational age and predict delivery time [27,28]. In a recent study, Pavan et al. evaluated B-mode ultrasonography and ARFI elastography of the central nervous system of canine fetuses as complementary methods to predict gestational age, monitor fetal development, and establish standards [29]. Ultrasound examinations were performed on English Bulldog bitches at 34, 49, and 60 days of gestation, measuring the circumference, area, and diameters of the short and long axis of the two cerebral hemispheres of the fetuses in cross-sections [29]. In dogs, the fetal cerebellum shape, echotexture, echogenicity, and transverse diameter were evaluated in cross-sections. Brain masses had a circular-to-oval shape, hyperechoic echogenicity, and homogeneous echotexture [29]. The cerebellum had a “banana” shape, with hyperechogenic edges, hypoechoic echogenicity, and homogeneous echotexture [29]. Modern high-resolution ultrasound images enable earlier assessment of measures of fetal development, including the identification of the bowel [30]. The ultrasonographic study of fetal bowel development represents a further method to correlate the organogenesis with gestational age [30]. Moreover, it could be interesting to establish a fetal age on femur length as in humans [20]. Unfortunately, in the present study, it was not possible due to the fetal position that did not allow a sufficient number of relevant measurements. Finally, monitoring the umbilical cord flow may be useful in order to establish an accurate index of fetus welfare, as it is already performed for other species [31,32].

## 5. Conclusions

Results of this study confirm that US can be used for early diagnoses of pregnancy by viewing the embryonic vesicle, embryo, and fetal development properly and fetal blubber thickness, heartbeat, and umbilical cord flow in California sea lion females. All these parameters showed a significant high correlation with gestational age and hence can be used for determining fetal age. This study provides additional relevant data on fetus–maternal ultrasonographic monitoring in sea lion species, in order to evaluate possible fetal distress or abnormalities. This information may be used as the basis for identifying gestational abnormalities in populations both under human care and in the wild. However, to better understand these findings, promptly diagnose congenital anomalies, and enhance the early detection of pregnancy complications, further studies are needed in a greater number of individuals to assess and validate reference baselines, and apply these diagnostic methods to other endangered seals and sea lion species. 

## Figures and Tables

**Figure 1 animals-14-01384-f001:**
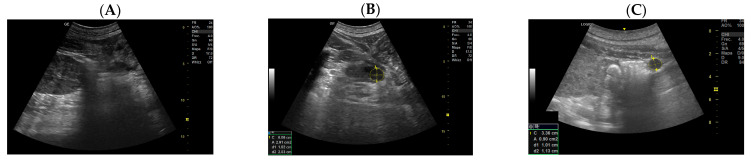
(**A**) Ovary with follicles, (**B**) dominant follicle, (**C**) CL with measurements.

**Figure 8 animals-14-01384-f008:**
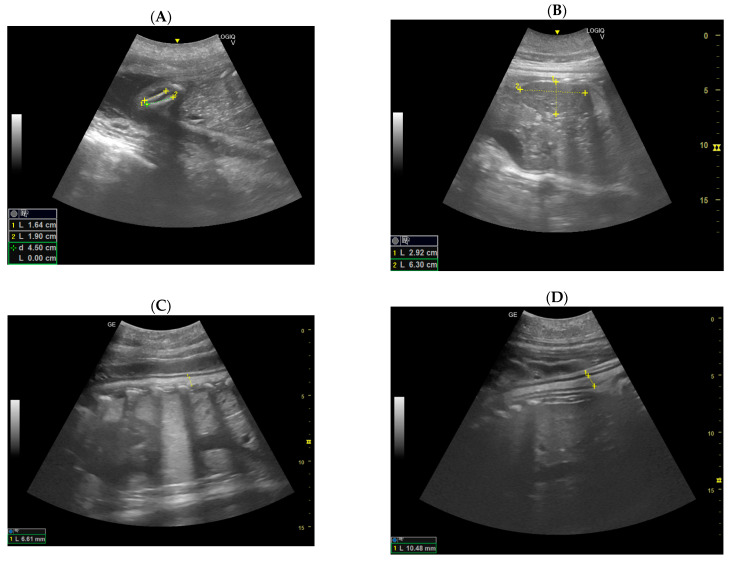
(**A**) Fetal femurs: A sonogram of the fetal femurs showing both the diaphysis and the cartilaginous epiphysis of the condyles and the major trochanter. The femur length (FL) is determined by placing the calipers at the junction between bone and cartilage; (**B**) kidneys with the measurements one month before the birth; (**C**) blubber; (**D**) it is of great interest to measure the blubber thickness to evaluate the weighted growth of the fetus.

**Figure 9 animals-14-01384-f009:**
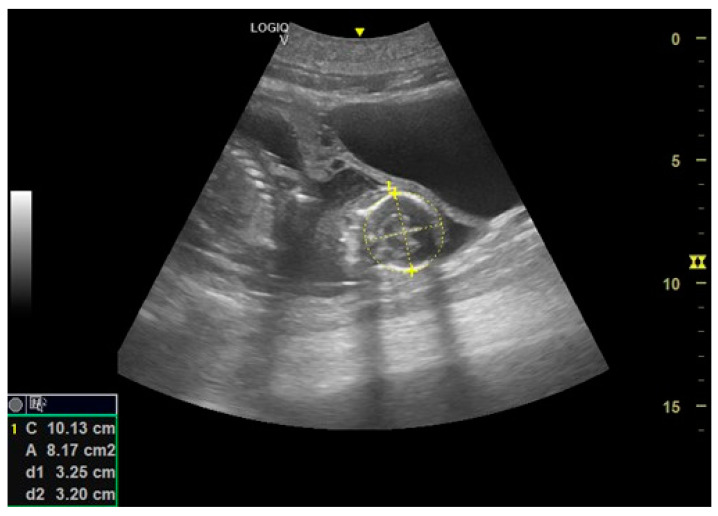
The US image shows a cross-section of the fetal head in a California sea lion at 201 days of gestation.

**Figure 10 animals-14-01384-f010:**
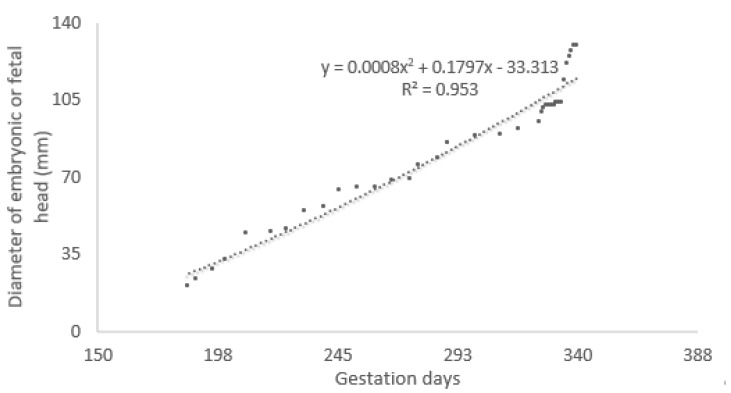
The growth curve of the diameter of the embryonic and fetal head during the course of gestation in the California sea lions.

**Figure 11 animals-14-01384-f011:**
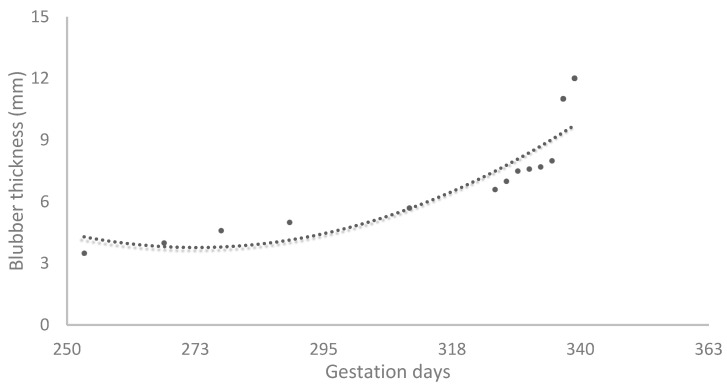
The growth curve of the blubber thickness during the course of gestation in the California sea lions.

**Figure 12 animals-14-01384-f012:**
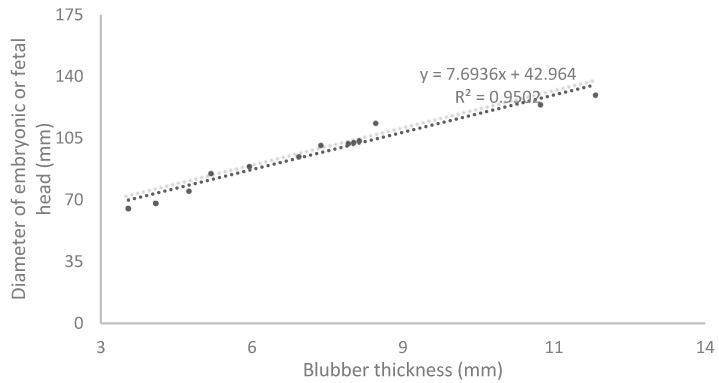
Correlation between the blubber thickness and the diameter of the embryonic and fetal head during the course of gestation in the California sea lions.

## Data Availability

Data are contained within the article.

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
