# Peer review of "Preliminary Ultrasonographic Study of Healthy California Sea Lion (Zalophus californianus) Pregnancy and Fetal Development"

_animals, 2024, doi:10.3390/ani14091384_

Round 1

Reviewer 1 Report

Comments and Suggestions for Authors

It was an interesting article to read; please consider the following edits. Thank you

I recommend including more literature in the introduction and discussion sections of the manuscript. I think the author should only talk about and discuss the parameters they study in this manuscript. Citations were not appropriate, and methods were not described thoroughly. Things/variables presented in the results section were not described in the methods section. I am not sure what statistical methods the author used and what the outcome and input variables were. Without an appropriate description of the methods section, this paper cannot pass journal rules. There was no line numbering in the manuscript. Which is very odd, and making a review report was difficult. Therefore, I am providing my review paragraph, page, and line-wise where appropriate.

Abstract line 5: What is 'yo'?

Abstract lines 8-9: no need to add detail of your ultrasound equipment. Just add that you did ultrasonography is enough.

Introduction lines 1-7: Please add the references.

Page 2: Line 7-16: You are presenting important numbers in the introduction but with no reference. Please cite appropriately.

Page 2, 2nd paragraph, line 6:    CL? Please add full terminology when you first use abbreviations.

Introduction, last paragraph, line 1:    US? Please add full terminology when you first use abbreviations.

M&M 1st paragraph: Animal collection or animal selection? which term would be more suitable?

M&M 1st paragraph: Again what is yo? US? Please add full terminology when you use any abbreviation first time.

In the method section of your report, you mentioned how you collected the data and which type of animals you used. However, I am curious to know what specific data you collected, such as the variables, how you analyzed them, what statistical methods or models you used, and what the input variables and outcome were. Without this information, it would be difficult to draw any conclusions or have a meaningful discussion about the results.

Section statistical analysis: What type of data you analysed and which type test/model you used to analysed the data?

Figure 10: Source of this p-value? what test you used?

Results section: variables you are presenting in results, you have not told the reader about these variables in methods section and how they were analyzed?

Author Response

Response to Reviewer 1 - Comments

Comments and Suggestions for Authors

It was an interesting article to read; please consider the following edits. Thank you
I recommend including more literature in the introduction and discussion sections of the ma- nuscript. I think the author should only talk about and discuss the parameters they study in this manuscript. Citations were not appropriate, and methods were not described thoroughly. Things/variables presented in the results section were not described in the methods section. I am not sure what statistical methods the author used and what the outcome and input varia- bles were. Without an appropriate description of the methods section, this paper cannot pass journal rules. There was no line numbering in the manuscript. Which is very odd, and making a review report was difficult. Therefore, I am providing my review paragraph, page, and line- wise where appropriate. Please, read the suggested changes in the main body of the manu- script.

Abstract line 5: What is ‘yo'? years old.
Abstract lines 8-9: no need to add detail of your ultrasound equipment. Just add that you did ultrasonography is enough. To the authors opinion, is mandatory to describe the model of the ultrasound machine and probe used, considering that all the ultrasound devices are different.
Introduction lines 1-7: Please add the references. DONE.
Page 2: Line 7-16: You are presenting important numbers in the introduction but with no reference. Please cite appropriately. DONE.
Page 2, 2nd paragraph, line 6: CL? Please add full terminology when you first use ab- breviations. Corpus luteum (CL). DONE.
Introduction, last paragraph, line 1: US? Please add full terminology when you first use abbreviations. Ultrasonography (US). DONE.
M&M 1st paragraph: Animal collection or animal selection? which term would be more suitable? Authors preferred to change “Animal collection” with “Study Animals”.
M&M 1st paragraph: Again what is yo? US? Please add full terminology when you use any abbreviation first time. DONE.
In the method section of your report, you mentioned how you collected the data and which type of animals you used. However, I am curious to know what specific data you collected, such as the variables, how you analyzed them, what statistical methods or models you used, and what the input variables and outcome were. Without this infor- mation, it would be difficult to draw any conclusions or have a meaningful discussion about the results. Section statistical analysis: What type of data you analysed and which type test/model you used to analysed the data?
Figure 10: Source of this p-value? what test you used?
Results section: variables you are presenting in results, you have not told the reader about these variables in methods section and how they were analyzed? The main goal of the study was to model the expected value of the diameter of the fetal head in terms of the

gestation age, the blubber thickness in terms of the gestation age and the blubber length in relation to the diameter of the fetal head.
Regression curves for these couple of variables was made using Medcalc software, Version 11.6.0.0, to analyze the data. A set of statistical processes for estimating the regressions pa- rameters and F-ratio significance level were determined. To describe a curve of correlation between fetal head diameter and gestation age, authors considered and realized the growth curve of the fetal head diameter. The major and minor axes of the maximum cross section of the embryonic and fetal heads were considered. The diameter of the fetal head was calculated as half the value of the sum of these two axes [25] (figure 9). From these results, authors estimated the following linear regression equation of the growth curve between the gestation age (the independent variable X) and the diameter of the fetal head (the dependent variable Y) (figure 10)

Y = 0.5882X – 86.4151,
(R2 = 0.9508, F-ratio significance level p < 0.0001).

Please, find in the main text the rest of changes, as required.

Reviewer 2 Report

Comments and Suggestions for Authors

Dear Authors,

You have to put a lot of work into this manuscript to make it suitable for publication.

Here You have my suggestion:

Brief report

The study Pregnancy and fetal development: ultrasonographic features from clinically healthy California sea lions (Zalophus californianus) under human care - presents an analysis of ultrasonographic monitoring of pregnancy in the title species. The study brings a lot of information to a previously undescribed species in terms of the parameters analysed - which is its main strength. I believe the paper should have been better contextualised and presented to make it more perfect and publishable.

The type of article suggests on 'short communication' - the number of animals is too low for me to call this article a calassic article.

Introduction

The introduction runs long strings of sentences without any citation. This is not up to the standard of article writing.

The research hypothesis is missing - In the introduction a lot of text is devoted to the analysis of embryonic development of the embryo, this is important, but at the same time some expected results of the ultrasound study are missing. Your team has experience in ultrasound examination of the fetus of aquatic animals, please write what results you expected.

Methodology

The procedure for ultrasound examination is strictly established. However, it is modified and personalised due to species variations. Please provide a protocol for such an examination (step-by-step). Please also demonstrate in the materials which parameters were examined in the context of the fetus and mebrional follicle.

Results

A breakdown of the results and how they differed between fetal sexes is missing.

Discussion

In my opinion, the discussion lacks a discussion of the information obtained with the development of the canid fetus. Sea lions belong to this group, but as we can see the development of this group of animals is significantly prolonged. What similarities can be seen during development? Which phase is the most elongated? Certainly the presence of the diapause phenomenon plays a role. This too has not been addressed in the discussion. 

The discussion is not just about repeating the results, the topic should be treated more extensively to give the reader an excellently developed topic. 

Conclusions

Line - "This study provides additional relevant data on fetus-maternal ultrasonographic monitoring in sea lion species, in order to evaluate possible fetal distress or abnormalities."

The results are sparse in data on the morphology and its changes of the reproductive organs in the female. Please supplement the results or reconstruct the conclusions so that they do not suggest that you have presented the changes that the sea lion uterus undergoes. 

Figures and tables

The figures you have posted are of good quality, however some are crooked and some have no scale applied, please correct this. 

I would like to suggest a table in which you represent the growth of the fetus on the basis of the parameters analysed, indicating the sex.

Detailed:

Line "Mating after a 28 days estrus yields a full year reproductive cycle, resulting in an twelve-month pregnancy with an approximate 90 days embryonic diapause." The cited paper is missing.

Line - "The embryo remains in this slow period of growth for 3 months until it is reactivated by maternal physiology." What physiology? Please quote and specify.

Line:

"3.1 Relationship between ultrasonographic image findings of embryonic and fetal growth and the age of gestation".

In this subsection the first "(In the US scan, sea lion ovary appears caudally to the kidney. Considering the 4 preg-nancies, the percentage of ovulation in the left ovary is 75%, while the ovulation in the right ovary was 25%. Maximum corpus luteum (CL) longitudinal diameter was 2.03cm (figure1)." sentences do not refer to embryo or fetal development. Please separate these data into a subsection, e.g. under the title Changes in the Maternal Reproductive System.

Line: "Day 129±3: embryonic cardiac mechanics is displayed, as a point of maximum fluctuation of the echoes." Was pulse or continuous Doppler used? Please also include this in the metrics and methods (state the range of flow velocity that was used for the flow function).

In addition, could the authors consider analysing the development of the heart, showing changes in heart wall thickness over the periods when these were visible?  

Line "In dog, previous studies confirmed a relationship between gestational age and fetal kidney growth. Measurement of fetal kidney length can be used in conjunction with other methods to estimate gestational age and predict delivery time [18,19,24]." Please also discuss other internal organs including the brain, heart among others.

Best regards

Author Response

Response to Reviewer 2 - Comments

Comments and Suggestions for Authors

Dear Authors,
You have to put a lot of work into this manuscript to make it suitable for publication. Here You have my suggestion:

Brief report

The study Pregnancy and fetal development: ultrasonographic features from clinically heal- thy California sea lions (Zalophus californianus) under human care - presents an analysis of ultrasonographic monitoring of pregnancy in the title species. The study brings a lot of in- formation to a previously undescribed species in terms of the parameters analysed - which is its main strength. I believe the paper should have been better contextualised and presented to make it more perfect and publishable. In this study, authors used ultrasonography to carry out a detailed observation of the embryonic and fetal growth process in pregnant California sea lion females under human care, considering that the mating date have been known, in order to establish a relationship between the embryonic and fetal development and the gestation age and to evaluate possible disorder or abnormalities. For this purpose, authors changed the title into: Preliminary ultrasonographic study of healthy California sea lions (Zalophus california- nus) pregnancy and fetal development. In fact, the study is preliminary, as there are no prece- dents in this species, and because the ultrasound study is purely descriptive, its main goal is to understand the possibilities that the ultrasound investigation has as a methodology for eva- luating feto-maternal well-being in this wild species as it has been done previously in dome- stic species.

The type of article suggests on 'short communication' - the number of animals is too low for me to call this article a classic article.
In order to follow what was suggested by both the Editor and the Reviewers, the Authors de- cided to change the type of manuscript from Article to Brief Communication

Introduction
The introduction runs long strings of sentences without any citation. This is not up to the standard of article writing. Please, see in the main text the references that authors added as suggested.
The research hypothesis is missing - In the introduction a lot of text is devoted to the analysis of embryonic development of the embryo, this is important, but at the same time some expected results of the ultrasound study are missing. Your team has experien- ce in ultrasound examination of the fetus of aquatic animals, please write what results you expected. In this study, authors used ultrasonography (US) to carry out a detailed obser- vation of the embryonic and fetal growth process in pregnant California sea lion females un- der human care, considering that the mating date have been known, in order to establish a relationship between the embryonic and fetal development and the gestation age and to eva- luate possible disorder or abnormalities. However, ultrasonography as some limitation in this

species, as the training in population under human care, or the pharmacological or physical restraint for the free-ranging once. Thus, this is a preliminary study, focused on the descrip- tion of the ultrasonographic findings during the sea lion pregnancy, that pretend to better un- derstand how the US investigation could be the diagnostic tool of choice for evaluating feto- maternal well-being in this wild species as it has been done previously in domestic species.

Methodology
The procedure for ultrasound examination is strictly established. However, it is modi- fied and personalised due to species variations. Please provide a protocol for such an examination (step-by-step). Please also demonstrate in the materials which parameters were examined in the context of the fetus and mebrional follicle. Please, read the sugge- sted changes in the main body of the manuscript.

Results
A breakdown of the results and how they differed between fetal sexes is missing.
Day 219±3 days: it also possible to recognize the eye as an anechoic cavitary structure (figure 6). During the last month of pregnancy is possible to identify most of the abdominal organs such as liver, spleen, intestine, kidneys easily (figure 7). In addition, the genitalia are visible and it could be possible to establish the sex of the fetus: in males is possible to identify the penis bone, which is obviously absent in females, but its visualization depends strongly on fetal position.

Discussion

In my opinion, the discussion lacks a discussion of the information obtained with the development of the canid fetus. Sea lions belong to this group, but as we can see the de- velopment of this group of animals is significantly prolonged. What similarities can be seen during development? Which phase is the most elongated? Certainly the presence of the diapause phenomenon plays a role. This too has not been addressed in the discus- sion.

Please, read the suggested changes in the main body of the manuscript.

The discussion is not just about repeating the results, the topic should be treated more extensively to give the reader an excellently developed topic.
Please, read the suggested changes in the main body of the manuscript.

Conclusions
Line - "This study provides additional relevant data on fetus-maternal ultrasonographic monitoring in sea lion species, in order to evaluate possible fetal distress or abnormali- ties."
The results are sparse in data on the morphology and its changes of the reproductive organs in the female. Please supplement the results or reconstruct the conclusions so that they do not suggest that you have presented the changes that the sea lion uterus undergoes. DONE.

Figures and tables
The figures you have posted are of good quality, however some are crooked and some have no scale applied, please correct this. DONE.

I would like to suggest a table in which you represent the growth of the fetus on the ba- sis of the parameters analysed, indicating the sex. Unfortunately, in the present study it is not applicable.

Detailed:

Line "Mating after a 28 days estrus yields a full year reproductive cycle, resulting in an twelve-month pregnancy with an approximate 90 days embryonic diapause." The cited paper is missing. ADDED.

Line - "The embryo remains in this slow period of growth for 3 months until it is reacti- vated by maternal physiology." What physiology? Please quote and specify. Modified as suggested.

Line:
"3.1 Relationship between ultrasonographic image findings of embryonic and fetal growth and the age of gestation".
In this subsection the first "(In the US scan, sea lion ovary appears caudally to the kid- ney. Considering the 4 preg-nancies, the percentage of ovulation in the left ovary is 75%, while the ovulation in the right ovary was 25%. Maximum corpus luteum (CL) longitudinal diameter was 2.03cm (figure1)." sentences do not refer to embryo or fetal development. Please separate these data into a subsection, e.g. under the title Changes in the Maternal Reproductive System. Modified as suggested.

Line: "Day 129±3: embryonic cardiac mechanics is displayed, as a point of maximum fluctuation of the echoes." Was pulse or continuous Doppler used? Please also include this in the metrics and methods (state the range of flow velocity that was used for the flow function). Pulse. DONE.

In addition, could the authors consider analysing the development of the heart, showing changes in heart wall thickness over the periods when these were visible? Unfortunately, in the present study it is not applicable.

Line "In dog, previous studies confirmed a relationship between gestational age and fe- tal kidney growth. Measurement of fetal kidney length can be used in conjunction with other methods to estimate gestational age and predict delivery time [18,19,24]." Please also discuss other internal organs including the brain, heart among other. Please, read the suggested changes in the main body of the manuscript.

Round 2

Reviewer 1 Report

Comments and Suggestions for Authors

I mostly agree with the edits made by the authors. However, simply writing "DONE" or "Please, read the suggested changes in the main body of the manuscript" in response to detailed comments is insufficient. The authors should include their response and the changes made to the manuscript in the rebuttal letter. Additionally, the authors should add line numbers to the manuscript to facilitate the review process. I previously pointed this out in the previous version of the manuscript, but it has not been addressed yet.

In regards to the statistical methods section, I believe that you have not fully explained the output and exposure variables. Although I can see what they are in the results, it is important to present the methods in detail in a scientific paper to ensure ease of understanding and adherence to scientific norms. I had previously mentioned this in the previous version of the manuscript, but it has not yet been added. Please add what were your output and exposure variables to the regression analysis.

I think regression is not a good option with a sample size of 2. However, the final decision rests with the Editor.

Author Response

Comments to Reviewer: We changed the manuscript as requested adding the missing literature both in the introduction and in the discussion. Also, we improved the methods section, fully explaining the output and variables in the statistical section and presenting the methods in detail. About the line numbering in the manuscript we added it as you can see.

Reviewer 2 Report

Comments and Suggestions for Authors

Thank you for all the corrections, now the article looks much better. 

Best regards

Author Response

Comments to Reviewer: thank you for your precious suggestion.